# Fabricating Well-Dispersed Poly(Vinylidene Fluoride)/Expanded Graphite Composites with High Thermal Conductivity by Melt Mixing with Maleic Anhydride Directly

**DOI:** 10.3390/polym15071747

**Published:** 2023-03-31

**Authors:** Jun Tong, Huannan Zheng, Jinwei Fan, Wei Li, Zhifeng Wang, Haichen Zhang, Yi Dai, Haichu Chen, Ziming Zhu

**Affiliations:** 1School of Mechatronic Engineering and Automation, Foshan University, Foshan 528000, China; 2School of Materials Science and Hydrogen Energy, Foshan University, Foshan 528000, China; 3Guangdong Key Laboratory for Hydrogen Energy, Foshan 528000, China; 4School of Education, City University of Macau, Macau 999078, China; 5Foshan Lepton Precision Measurement and Control Technology Co., Ltd., Foshan 528000, China; 6College of Life Science and Technology, Jinan University, Guangzhou 519070, China

**Keywords:** polymer–matrix composites (PMCs), microstructures, thermal properties

## Abstract

Maleic anhydride (MA) is introduced to fabricate poly(vinylidene fluoride)/expanded graphite (PVDF/EG) composites via one-step melt mixing. SEM micrographs and WAXD results have demonstrated that the addition of MA helps to exfoliate and disperse the EG well in the PVDF matrix by promoting the mobility of PVDF molecular chains and enhancing the interfacial adhesion between the EG layers and the PVDF. Thus, much higher thermal conductivities are obtained for the PVDF/MA/EG composites compared to the PVDF/EG composites that are lacking MA. For instance, The PVDF/MA/EG composite prepared with a mass ratio of 93:14:7 exhibits a high thermal conductivity of up to 0.73 W/mK. It is 32.7% higher than the thermal conductivity of the PVDF/EG composite that is prepared with a mass ratio of 93:7. Moreover, the introduction of MA leads to an increased melting peak temperature and crystallinity due to an increased nucleation site provided by the uniformly dispersed EG in the PVDF matrix. This study provides an efficient preparation method for PVDF/EG composites with a high thermal conductivity.

## 1. Introduction

Thermal conductive materials exhibit great potential in electronic devices, heat exchangers, and automobiles. Products that come from materials with a high thermal conductivity can maintain them at relatively low temperatures by rapidly diffusing the heat, which is very important for extending the service life of products and lowering the risks [1], whereas most polymers experience difficulties when meeting the applications due to their thermal insulating properties. For example, poly(vinylidene fluoride) (PVDF), which is known for its potential application in the field of engineering technology based on its mechanical and electrical properties, exhibits low thermal conductivity due to its low glass transition temperature. Improving the thermal property of PVDF by adding fillers with an excellent thermal conductivity has been extensively investigated. Conductive allotropes of carbon [2,3,4,5,6] are potential candidates for enhancing the conductive properties of polymers, which have attracted a lot of attention from researchers. For example, expanded graphite (EG), which can be prepared by rapidly heating graphite intercalation compounds (GIC) [7,8], is deemed to be a valid candidate for the conductivity improvement of polymers due to its high conductivity. However, EG can be aggregated and restacked easily in polymer matrices owing to the van der Waals forces among them and the large differences in surface energy between graphite and most polymers, which seriously restrict the conductivity of the composites [9,10,11,12]. So, the appropriate surface functionalization of EG layers is normally required to facilitate their dispersion in the polymer matrix by enhancing the interfacial adhesion between them [13].

Song et al. [14] prepared epoxy/graphene nanocomposites via the solution mixing method with the introduction of non-covalent functionalized graphene. The thermal conductivity of the composites filled with functionalized graphene increased significantly due to the increased conduction paths for the phonons in the epoxy matrix caused by the improved dispersion of graphene. Xu et al. [15] prepared polyvinylidene fluoride (PVDF)/graphene composites via the solution-cast method, in which the graphene was modified by mechanical stirring with ionic liquid (IL) in acetone at 80 °C for 2 h. As a result, the dispersion of graphene in the PVDF obviously improved. Zhang et al. [16] modified EG previously by dispersing it in a NaOH solution with the introduction of IL via ultrasonic and then mixing the solution with an epoxy. The EG dispersion state obviously improved, and the composites exhibited superior electrical and thermal conductivities.

The modifications of graphene or EG mentioned above were usually implemented in solutions by mechanical stirring or ultrasonication, and most of them asked for several processing steps and/or relatively demanding conditions, which might give rise to environmental pollution and low yields. Graphene modification methods with mild conditions were developed based on the Diels–Alder reaction, which proverbially took place between dienophiles and conjugated dienes. In these methods, graphene/graphite generally acted as a diene [17,18,19] and they could be covalently attached to alkene compounds (dienophile species) [18,20,21,22], such as maleic anhydride (MA), via the Diels–Alder cycloaddition. The theoretical calculation and experimental results proposed by Zabihi et al. [23] demonstrated that MA could attach onto the active sites of a graphite platelet by a high collision force at its edge, where the electrons were abundant.

MA is a polar compound that is commonly used to functionalize non-polar polymers to enhance their interaction with non-polar polymers and fillers [24,25,26,27]. Recently, MA was also used for polar PVDF preparation due to the reactivity of the anhydride group on the MA monomer [28]. The research results demonstrated that the carbonyl groups of the MA could exhibit a strong interaction with the methylene groups of the PVDF during melt mixing.

Thus, in this work, MA was chosen as a modifying agent to facilitate EG dispersion in the matrix of PVDF by enhancing the interfacial adhesion between the EG and the PVDF. It is a well-known fact that melt mixing is an economically and green method for the preparation of polymer/filler composites due to the feasibility of scalable fabrication and the absence of solvents [12,29]. So, one-step melt mixing was carried out to prepare PVDF-based composites with the direct introduction of MA, which would improve the dispersion of EG in the PVDF matrix in an efficient way. The thermal conductivity of the composites and the crystallization behavior and crystal structure of PVDF in the prepared samples were tested. Furthermore, the mechanism for EG dispersion in the PVDF matrix during melt mixing was analyzed in detail. This study aims to expand the application of PVDF in the field of electronic devices, where high thermal conductivity is needed [30].

## 2. Materials and Methods

### 2.1. Materials

GIC (grade XF057X) was supplied by Nanjing XFNANO Materials Tech Co., Ltd., Nanjing, China. The temperature for initial expansion of GIC is 200 °C. MA with a content higher than 99.5% was supplied Tianjin Zhiyuan Chemical Reagent Co., Ltd., Tianjin, China. PVDF (grade FR 906) with a melt flow index of 18.0 g/10 min (at 230 °C and 5 kg) and a density of 1.79 g/cm^3^ was supplied by Shanghai 3F New Materials Co., Ltd., Shanghai, China.

### 2.2. Fabrication of Composites

Internal mixer (HTK-300, Hartek Tech Co., Ltd., Guangzhou, China) equipped with roller as the rotor was used to prepared PVDF/MA/EG composites by melt mixing. Before melt mixing, PVDF was first vacuum-dried for four hours at 80 degrees Celsius. The PVDF pellets were mixed in the internal mixer at conditions of 100 rpm and 220 °C until they were fully molten (about 1 min). Then, MA/GIC mixture, which was premixed with a mass ratio of 2:1, was added into the PVDF melt. Then, they were mixed in the internal mixer until there were no gases generated due to the expansion of GIC (about 10 min). The PVDF/MA/EG composites prepared with mass ratios of 97:6:3 and 93:14:7 were denoted as PM6G3 and PM14G7, respectively. For comparison, PVDF/EG composites with mass ratios of 97:3 and 93:7 were also prepared and denoted as PG3 and PG7, respectively. Moreover, the PVDF/MA blends were prepared with mass ratios of 100:6 and 100:14 for comparison, and denoted as PM6 and PM14, respectively. Then, all the prepared samples were hot-pressed at 220 °C. Specimens used for characterization were taken from the molded samples.

### 2.3. Characterization

Microstructure of the composites was characterized using SEM (EM 30N, Coxem, Daejeon, Korea). All the specimens used for SEM were gold-sputtered after being cryofractured and then they were tested at an accelerating voltage of 15 kV.

WAXD tests were implemented on an X-ray diffractometer (D8 Advance, Bruker, Germany) at 2*θ* angle at 2−40°, with a scanning step of 0.01° and scanning rate of 2°/min. The specimens used for test were 10 mm in wide and 1 mm in thickness.

FTIR tests were achieved on a Vertex 70 spectrometer (Bruker, Germany) equipped with an attenuated total reflectance (ATR) cell. Specimens sized 10 × 10 × 1 mm^3^ were tested at scanning range from 4000 to 400 cm^−1^ and resolutions of 4 cm^−1^. Each specimen was tested thrice.

Hot Disk TPS 2500S (Hot Disk, Uppsala, Sweden) was used as a thermal constant analyzer to test the samples’ thermal conductivities in thickness direction at 30 °C. Specimens used for thermal conductivity measurement were 25 mm (diameter) × 2 mm (thickness) in size. Three specimens were measured for each sample.

Crystallization behavior of the samples was tested on a DSC (Netzsch STA 449 F5, Selb, Germany) for crystallization and melting behaviors characterization. The specimens were heated to 220 °C, cooled to 30 °C, and heated to 220 °C again by 10 °C/min during testing. Significantly, thermal histories of the samples were erased by keeping the samples at 220 °C for 5 min before cooling. Crystallinities (*X*_c_s) of the samples were calculated by Equation (1):(1)$$X_{c}=\frac{{∆H}_{m}}{{∆H}_{m}^{0}\times \omega }\times 100\%$$
where Δ*H*_m_ and Δ$H_{m}^{0}$ are melting enthalpies of samples and 100% crystalline polymer, respectively, and *ω* is the mass fraction of PVDF in the prepared samples. The Δ$H_{m}^{0}$ of PVDF is 104.6 J/g [31].

Dynamic rheology properties were tested at 220 °C by Anton Paar Physica MCR 302. Specimens that were 25 mm in diameter and 1 mm in thickness were measured at 1.0% strain with frequency sweeps ranging from 0.01 to 100 Hz.

Thermogravimetric analysis for MA was carried out on thermogravimetric analyzer (Netzsch STA 449 F5, Selb, Germany). Temperature for test ranged from 30 to 200 °C and increased by 10 °C/min under nitrogen atmosphere.

## 3. Results and Discussion

### 3.1. Dispersion State of EG

The dispersion states of EG in the PVDF matrix for all the prepared composites were tested by SEM and the results are illustrated in Figure 1. As can be seen, some fluffy EG aggregates, which were derived from the thermal expansion of GIC, are displayed in the PVDF matrix for both composites prepared with and without MA addition. Under the action of screw shear, some EG layers are exfoliated from the aggregates and are dispersed in the PVDF matrix. Compared to the PVDF/EG composites (Figure 1a,c), the PVDF/MA/EG composites exhibit much smaller aggregates, as illustrated in Figure 1b and d, which means that the introduction of MA facilitates the dispersion of EG in the PVDF matrix.

Evidence can also be found from the WAXD patterns of the PVDF/EG and PVDF/MA/EG composites samples. As presented in Figure 2, a peak appears at about 2*θ* = 26.5° in all the composites, which corresponds to the characteristic peak of EG [32]. With the increase in the EG content, the peak intensity in the PVDF/EG composites sample increases, whereas with the addition of MA, the PM6G3 and the PM14G7 samples exhibit a much lower intensity of the EG characteristic peak than the PG3 and the PG7 samples, respectively. Previous research [32,33,34] has shown that a lower filler characteristic peak intensity would present at the spectrograms of composites with a well-dispersed state. The decreased intensity of the EG characteristic peak for the PVDF/MA/EG composites in this work further confirms that a better dispersion state of EG is obtained in the composites that were prepared with the addition of MA.

### 3.2. Crystallization Behavior and Crystal Structure

The crystallization behavior of PVDF was characterized by DSC. The DSC cooling and heating curves for the PVDF, PG3, PM6G3, PM6, PG7, PM14G7, and PM14 samples are presented in Figure 3. As illustrated in Figure 3a, the PVDF sample exhibits a crystallization peak temperature (*T*_c_) at about 133.8 °C. With the addition of EG, the *T*_c_ increases to 134.2 and 135.5 °C along with the narrowed peak band for the PG3 and PG7 samples, respectively. This indicates that the EG acts as a seed for heterogeneous nucleation, accelerating the crystallization of PVDF. With the addition of MA, the *T*_c_ of the PM6G3 and the PM14G7 samples decreases to 133.2 and 134.5 °C, respectively. This can be attributed to the diluting effects of MA on the PVDF matrix, leading to the decelerated crystallization of PVDF [28]. Figure 3b shows the DSC heating curves for the prepared samples. The *X*_c_s of the prepared samples is calculated using Equation (1), and the parameters derive from the DSC heating curves. As listed in Table 1, the melting peak temperature (*T*_m_) and the *X*_c_ of the PVDF sample are 175.8 °C and 21.0%, respectively. With the addition of EG, the *T*_m_ and the *X*_c_ of the composites decrease as the EG content increases. As can be seen, the *T*_m_ and the *X*_c_ of the PG3 sample are 175.1 °C and 20.4%, respectively, while the *T*_m_ and the *X*_c_ of the PG7 sample further decrease by 173.7 °C and 19.9%, respectively. This can be ascribed to the fact that the introduced EG with a large specific surface area restrains the movement of the PVDF molecular chains, impeding the growth of the crystal, and leading to an imperfect crystal structure, whereas with the cooperation of MA, the PVDF/MA/EG composites exhibit higher *T*_m_ and *X*_c_ than the PVDF/EG composites, which can be attributed to the well-dispersed EG facilitated by MA, leading to increased nucleation sites in the PVDF matrix.

The polymorphisms of the prepared samples were characterized by FTIR and WAXD. The typical FTIR spectra for the MA, PVDF, PM6, PM14, PG3, PG7, PM6G3, and PM14G7 samples are displayed in Figure 4. The PVDF sample exhibits transmittance peaks at 763, 794, and 976 cm^−1^, which refer to the *α*-phase (TGTG′ conformer) of PVDF. Moreover, three weak transmittance peaks arise at 510, 840, and 1274 cm^−1^. The peaks at 510 and 840 cm^−1^ correspond to both polar *β*- and *γ*-phases (TTTT and TTTG conformers) [31]. Additionally, the transmittance peak at 1274 cm^−1^ refers exclusively to the *β*-phase [35,36]. With the addition of EG, the transmittance peaks referring to *α*- and *β*-phases for the PVDF/EG composites exhibit no obvious change. However, for the PVDF/MA samples, the transmittance peak corresponding to the *β*-phase, especially at 1274 cm^−1^, shows a weak intensity, meaning that the MA in the PVDF matrix restrains the formation of *β* crystals.

Figure 5 illustrates the WAXD spectra for the PVDF, PM6, PM14, PG3, PG7, PM6G3, and PM14G7 samples. The PVDF sample exhibits three characteristic diffraction peaks assigning to (100), (020), and (110) reflections of the *α*-phase crystals of PVDF at 2*θ* = 17.7, 18.4, and 20.0°. With the introduction of EG, especially for the PG7 sample, the three characteristic peaks shift to 2*θ* = 17.8, 18.5, and 20.1°, corresponding to the *γ*-phase crystals of PVDF [37]. With the addition of MA, the three characteristic peaks show no obvious change compared to these of the PVDF sample, which implies that the added MA suppresses the formation of the polar crystals of PVDF.

### 3.3. Thermal Conductivity

Figure 6 illustrates the thermal conductivities of the PVDF, PM6, PM14, PG3, PM6G3, PG7, and PM14G7 samples coupled with standard deviations. The thermal conductivity of the PVDF sample is about 0.22 W/mK. With the addition of EG, the thermal conductivities increase to about 0.36 and 0.55 W/mK for the PG3 and PG7 samples, respectively, which is mainly ascribed to the high thermal conductivity of the EG dispersed in the PVDF matrix. Moreover, the introduction of MA may also increase the thermal conductivity of the PVDF to some extent by increasing its *X*_c_ (as listed in Table 1), which can effectively facilitate thermal diffusion by reducing the interfacial phonon scattering [38,39]. For the PVDF/EG composites prepared with the addition of MA, they possess much higher thermal conductivities than the composites prepared without the addition of MA. The thermal conductivities of the PM6G3 (0.44 W/mK) and PM14G7 (0.73 W/mK) samples are 22.2% and 32.7% higher than those of the PG3 and PG7 samples, respectively. It is mainly ascribed to the improved EG dispersion state in the PVDF matrix (as illustrated in Figure 1). Although, Deng et al. [40] successfully prepared PVDF/EG composites with a high thermal conductivity of 1.29 W/mK by introducing 15 wt% EG. The EG was treated via ball milling for more than 10 h before the EG was mixed with the PVDF. The method proposed in this work exhibits high efficiency for the preparation of PVDF/EG composites with high thermal conductivity.

### 3.4. Mechanism for EG Dispersion during Melt Mixing

From the foregoing discussion, the PVDF/EG composites with a higher thermal conductivity are actually fabricated with the introduction of MA in this work due to the improved dispersion of EG. Figure 7 schematically shows the mechanism for EG dispersion during melt mixing and this is analyzed in detail as follows.

Generally, the GIC will expand and generate EG if the temperature is higher than its initial expansion temperature (about 200 °C). In this work, GIC powder is introduced directly into the PVDF melt while the internal mixer is working at a rotor speed of 100 r/min and a temperature of 220 °C. During the melt mixing, quantities of gases (such as water vapor and carbon dioxide) are released from the graphite interlamination at high pressure due to the decomposition of oxidized groups in the GICs, which can expand the GICs and generate EG with large interlamellar spacing [32,41]. Therefore, the EG layers can be intercalated by the PVDF molecular chains and then exfoliated and dispersed under the rotor shear during melt mixing. However, the dispersibility of EG is limited due to the high viscosity of the PVDF melt and the poor interaction between the PVDF and the EG, which disadvantages the PVDF molecular chains in terms of intercalating into the graphite interlamination of EG.

For the PVDF/MA/EG composites, the introduction of MA can not only facilitate the mobility of the PVDF molecular chains, but it can also enhance the interaction between the PVDF and the EG. Figure 8 illustrates the complex viscosity (*η**) versus frequency for the PVDF, PM6, PG3, PM6G3, PM14, PG7, and PM14G7 samples. The PVDF/EG composite samples present a higher *η** than the PVDF sample, especially at low frequencies. With the addition of MA, the *η**s of the samples obviously decrease, which is contrary to the findings of the research reported by Ye et al. [28], which state that the PVDF/MA samples possess a higher *η** than the PVDF sample due to the strong interaction between PVDF and MA, whereas it is also reported that there is a competitive relation between interaction and plasticization with the addition of MA. The effect of plasticization on the PVDF may be stronger than the effect of the interaction with PVDF under the conditions in this work. So, the introduced MA can promote the intercalation of EG by facilitating the mobility of PVDF molecular chains during melt mixing.

The interaction between the MA and the PVDF can be demonstrated by the TGA result of the MA and the FTIR spectra of the prepared samples. As illustrated in Figure 9, the neat MA shows poor thermal stability, which can be almost completely degraded at a temperature of 183 °C, whereas comparing to the PVDF sample, several new characteristic peaks referring to the MA are presented in the FTIR spectra of the PVDF/MA blends and PVDF/MA/EG composites samples, especially those with a high MA content experiencing melt mixing at 220 °C (as can be seen in Figure 10a). This means that the thermal stability of MA in the PVDF/MA blend and PVDF/MA/EG composites is enhanced, which confirms the strong interaction between the MA and the PVDF. Furthermore, the PM14 and the PM14G7 samples exhibit an obvious symmetric stretching band of the carbonyl group (C=O) of MA at 1781 cm^−1^ [42,43,44], which is red-shifted relative to that of the neat MA (1774 cm^−1^). Simultaneously, the PVDF, PG3, and PG7 samples exhibit a deformed vibration of the CH_2_ groups of PVDF chains at 1382 cm^−1^, as can be seen in Figure 10b. With the addition of MA, the characteristic peak at 1382 cm^−1^ shifts to 1380 cm^−1^. According to the results above, it can be inferred that the shifts result from the hydrogen bonding interaction between the CH_2_ group of the PVDF and the C=O group of the MA (as illustrated in Figure 7).

The interaction between the MA and the EG cannot be demonstrated by the results mentioned above, which is directly due to the low content of them in the samples, whereas it has been confirmed by the density functional theory and computational and experimental methods that electron flow toward the graphite edge derived from the un-paired π electrons on the graphite plane can be induced by the high-energy collisions [23,41]. So, a negative charge might also be formed locally at the graphite plane edge when a collision of EG occurs during melt mixing in this work. Then, the MA, which acts as a dienophile, covalently reacts with the active sites at the graphite edges by forming sigma bonds, as shown in Figure 7. Therefore, the interaction between the PVDF and the EG is enhanced with the addition of MA, which acts as a binder.

Above all, the EG dispersion state is significantly improved due to the introduction of MA by facilitating the PVDF molecular chains’ mobility and by strengthening the interaction between the EG and the PVDF. As a result, the mean inter-layer spacing of EG in the PVDF matrix is reduced, which improves the thermal conductivities of the composites by facilitating the phonon transport [45].

## 4. Conclusions

High thermal conductivity is obtained for the PVDF/MA/EG composites due to the improved EG dispersion state in the PVDF matrix, which has been demonstrated by the SEM and WAXD diffraction results of the composites. On the one hand, the introduced MA can promote the intercalation of EG by facilitating the mobility of PVDF molecular chains during melt mixing due to the plasticization of MA. On the other hand, the added MA can enhance the interfacial adhesion between the PVDF and the EG by forming hydrogen bonds with the PVDF and sigma bonds with the graphite edges simultaneously. With the addition of MA, the thermal conductivities of the PM6G3 (0.44 W/mK) and PM14G7 (0.73 W/mK) samples are 22.2% and 32.7% higher than those of the PG3 and PG7 samples, respectively. Although, the added MA may decelerate the crystallization of the PVDF due to the diluting effects of MA on the PVDF matrix. The *T*_m_ and *X*_c_ of the PVDF/MA/EG composites are higher than those of the PVDF/EG composites, which can be attributed to an increased nucleation site caused by the improved EG dispersion state, whereas the addition of MA would suppress the formation of polar crystals of PVDF. In summary, an efficient method for PVDF/EG composites with a high thermal conductivity by improving the dispersion of EG through melt mixing with maleic anhydride directly is proposed in this work, which would provide more possibilities for expanding the application of PVDF in the field of engineering technology.

## Figures and Tables

**Figure 1 polymers-15-01747-f001:**
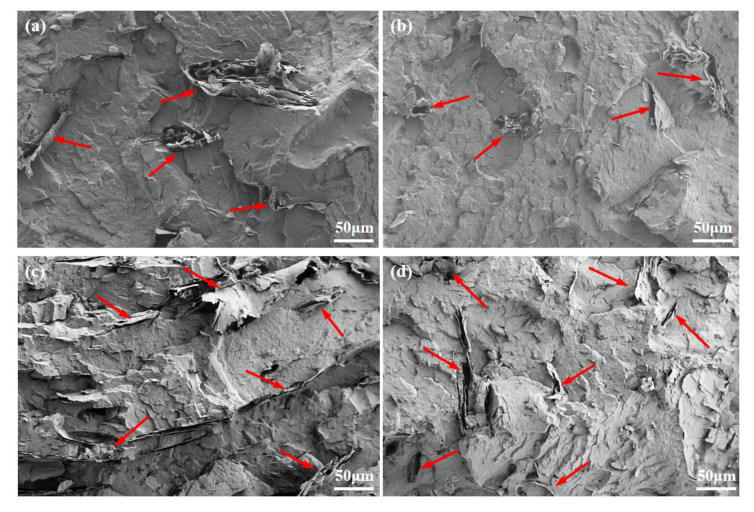
SEM micrographs of (**a**) PG3, (**b**) PM6G3, (**c**) PG7, and (**d**) PM14G7 samples.

**Figure 2 polymers-15-01747-f002:**
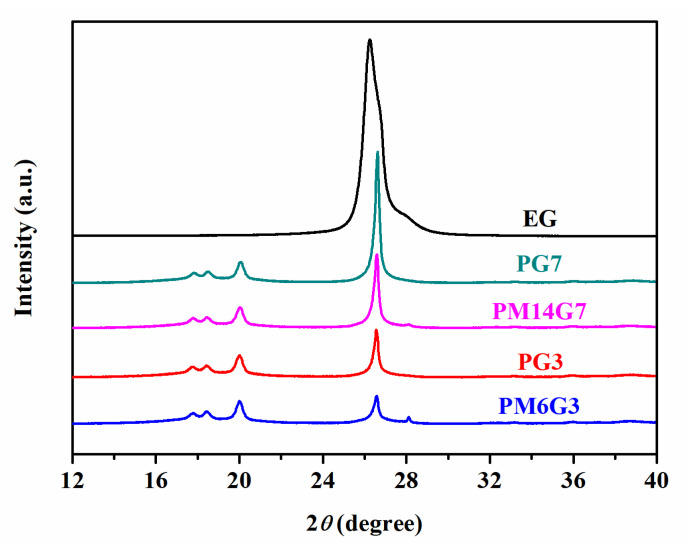
WAXD spectra of PG3, PM6G3, PG7, and PM14G7 samples.

**Figure 3 polymers-15-01747-f003:**
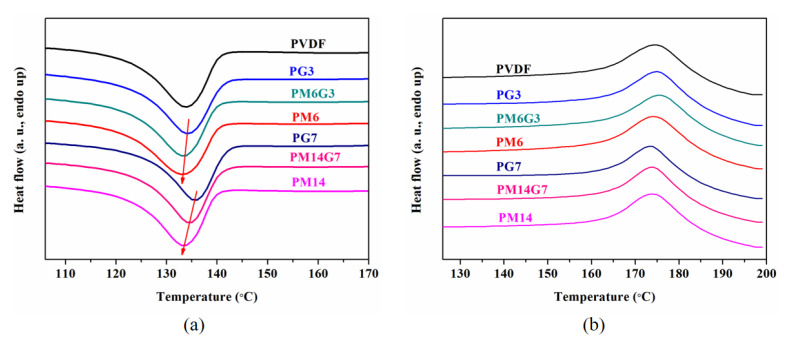
DSC (**a**) cooling and (**b**) heating curves for PVDF, PG3, PM6G3, PM6, PG7, PM14G7, and PM14 samples.

**Figure 4 polymers-15-01747-f004:**
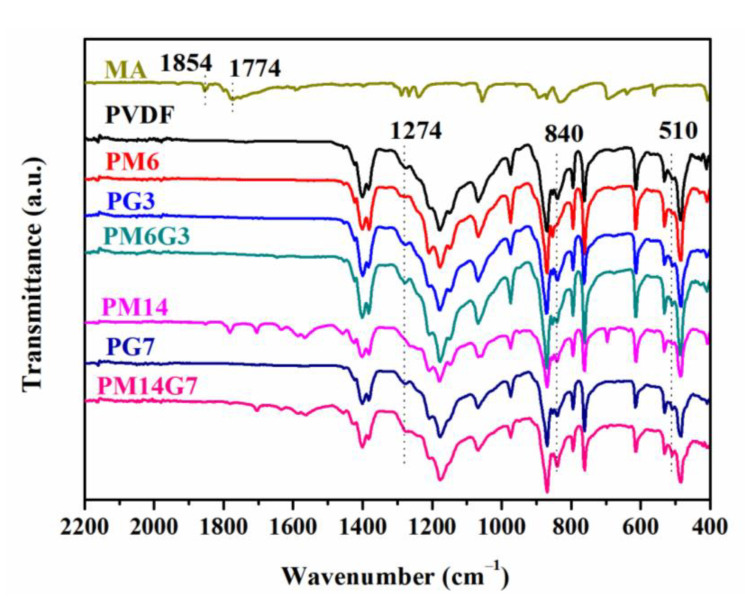
FTIR spectra of MA and all prepared samples.

**Figure 5 polymers-15-01747-f005:**
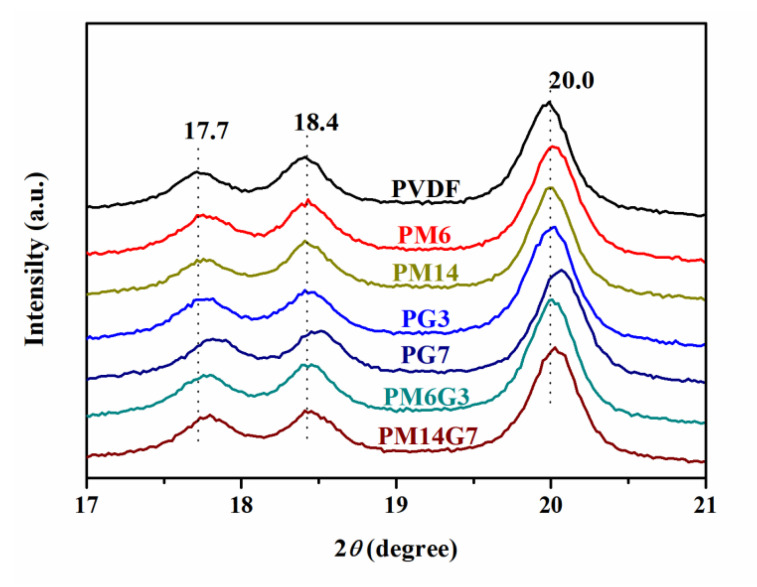
WAXD spectra of PVDF, PM6, PM14, PG3, PG7, PM6G3, and PM14G7 samples.

**Figure 6 polymers-15-01747-f006:**
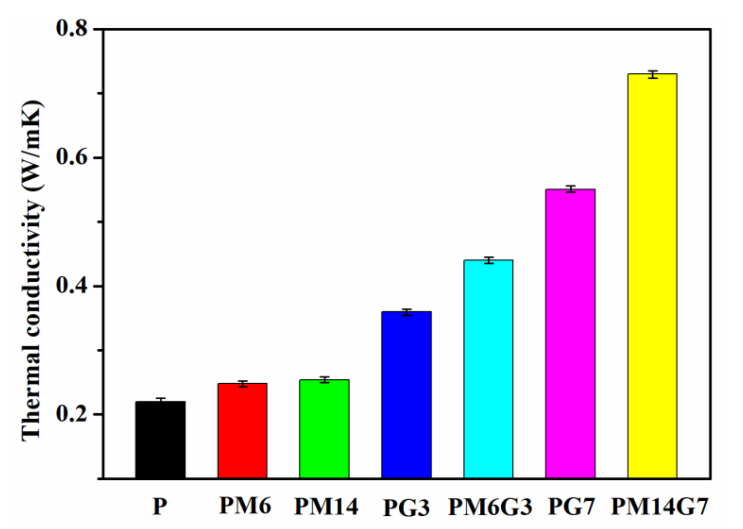
Thermal conductivities of PVDF, PM6, PM14, PG3, PM6G3, PG7, and PM14G7 samples.

**Figure 7 polymers-15-01747-f007:**
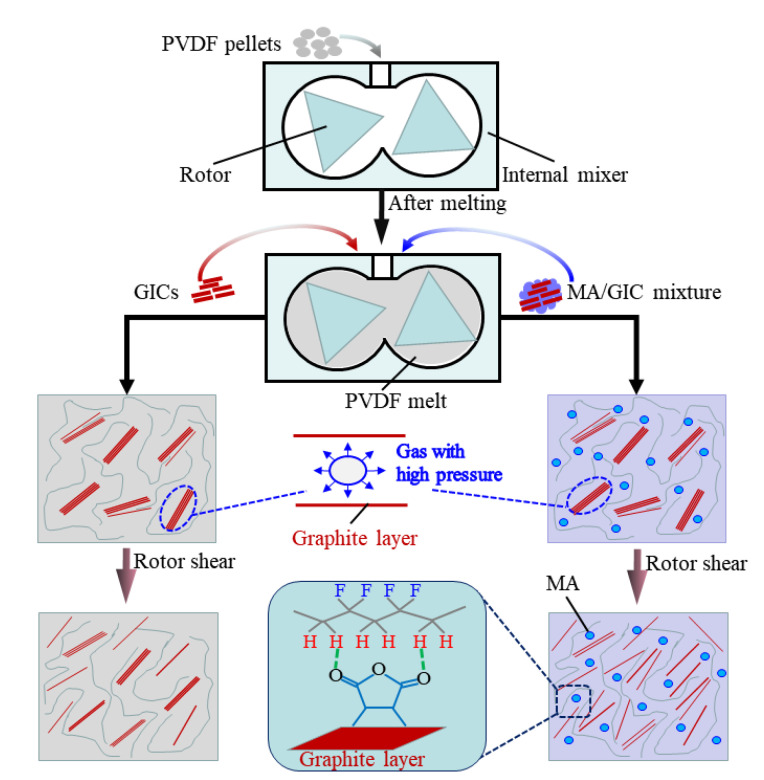
Schematics of mechanism for dispersion of EG in PVDF matrix during melt mixing with the introduction of MA.

**Figure 8 polymers-15-01747-f008:**
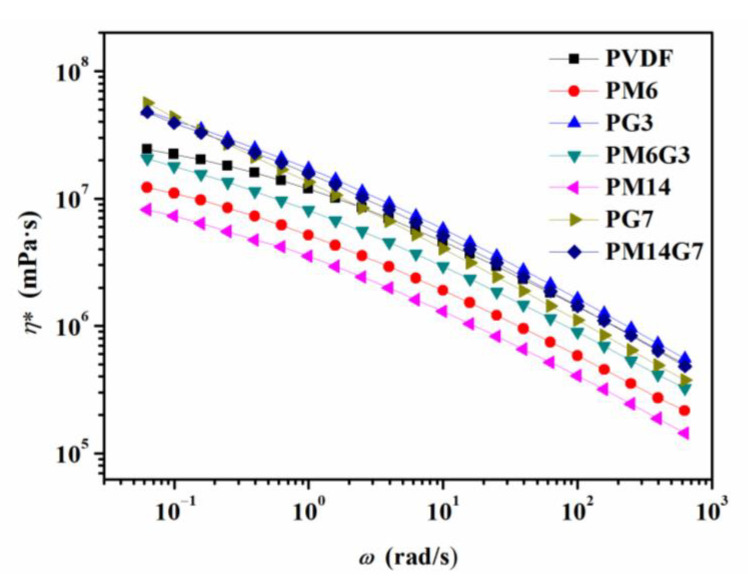
Complex viscosity versus frequency curves for PVDF, PM6, PG3, PM6G3, PM14, PG7, and PM14G7 samples.

**Figure 9 polymers-15-01747-f009:**
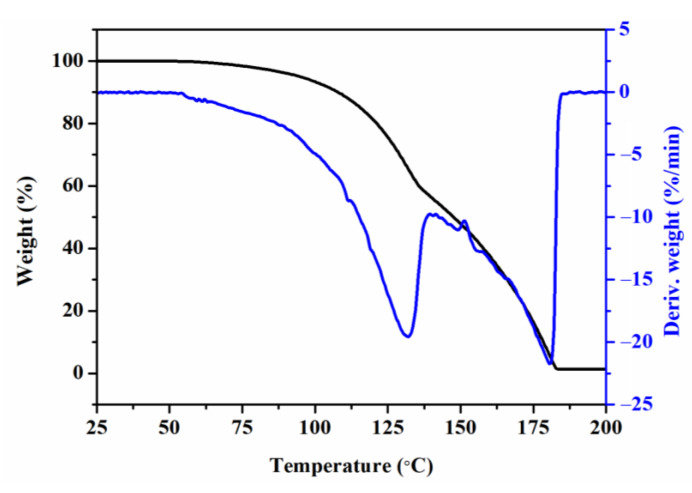
TGA and DTG thermograms for MA.

**Figure 10 polymers-15-01747-f010:**
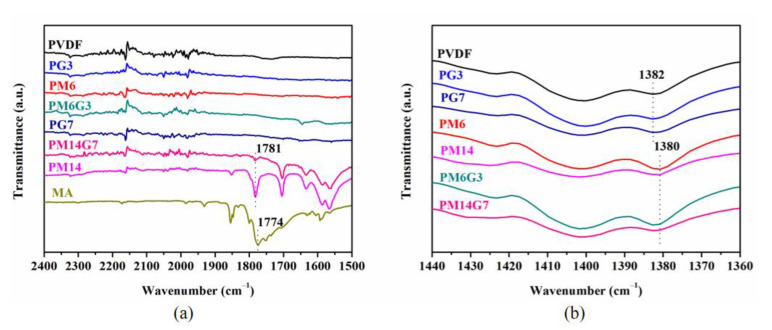
(**a**) C=O symmetric stretching band of MA and (**b**) CH_2_-deformed vibration of PVDF in the prepared samples.

**Table 1 polymers-15-01747-t001:** DSC thermal performance parameters for PVDF, PG3, PM6G3, PM6, PG7, PM14G7, and PM14 samples.

Sample	*T*_c_ (°C)	*T*_m_ (°C)	*H*_m_ (J/g)	*X*_c_ (%)
PVDF	133.8	175.8	21.97	21.0
PG3	134.2	175.1	20.71	20.4
PM6G3	133.2	176.0	19.89	20.7
PM6	132.9	174.6	22.71	23.0
PG7	135.5	173.7	19.50	19.9
PM14G7	134.5	174.1	20.36	23.6
PM14	133.4	174.2	21.36	23.3

## Data Availability

No new data were created.

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
