# Peer review of "Fabricating Well-Dispersed Poly(Vinylidene Fluoride)/Expanded Graphite Composites with High Thermal Conductivity by Melt Mixing with Maleic Anhydride Directly"

_polymers, 2023, doi:10.3390/polym15071747_

Round 1

Reviewer 1 Report

This work reports on the covalent functionalization of expanded graphite (GIC) with malice acid (MA) via Diels-Alder reaction in a melt with PVDF polymer. The composites containing MA exhibited better dispersion of GIC flakes, and had increased thermal conductivity and stability compared to composites formed solely with GIC and PVDF. Materials were characterized by electron microscopy, XRD, TGA, DSC and FTIR. Results and materials are very interesting and should appeal to a broad audience of scientists and engineers working with thermal conductivity of composites for numerous applications, such as energy storage and conversion devices, electronics packaging, to name a few. I have only some recommendations for authors in order to improve their discussion and more strongly support their conclusions. My comments are:

1. Language requires improvement throughout the text, more specifically on consistence and proper use of the past perfect tense.

2. Introduction should cite some papers that require specifically PVDF composites with good thermal conductivity, or explain why PVDF was selected for the study.

3. Figure 9: TGA for MA should be compared to the TGA for GIC, PVDF, and for one composite of GIC/MA/PVDF and one of GIC/PVDF. In this way authors may be able to discuss the changes in the structure of GIC, as well as any improvements in the thermal stability of composites.

Reviewer 2 Report

Dear,

The authors compounded PVDF/EG with maleic anhydride to increase dispersion efficiency. The central theme was to improve the thermal conductivity of the compounds. In my opinion, the authors should add the results of the expanded graphite characterization. The manuscript needs to be revised, as highlighted below:

> Abstract. Authors must present the conclusion at the end of the abstract, informing the importance of the manuscript;

> Introduction. Please make clear the novelty of the manuscript, as well as the importance of the study for scientific advancement; Please update manuscript references;

> Page 3. “The PVDF/MA/EG composites prepared with mass ratios of 97:6:3 and 93:14:7”. It does not close 100% mass. Is there any material used in resin hundred parts (phr)? Please clarify; Also check: PVDF/MA blends were prepared with mass ratios of 100:6 and 100:14 for comparison

> FTIR. Add the scan amount and sample type (powder; film; thick sample);

> Why did the authors not develop composites with expanded and non-expanded graphite? This way you would have a better idea of the effect of graphite expansion;

> Authors should add expanded graphite characterizations in the manuscript for comparative purposes;

> Page 4. “characteristic peak for the PVDF/MA/EG composites in this work further confirms that the better EG....”. Why did the authors not perform transmission microscopy (TEM) to better assess dispersion?

> Page 5. “Whereas, with the cooperation of MA, the PVDF/MA/EG composites exhibit higher Tm and Xc than the PVDF/EG.......”. The maleic anhydride is acting as a nucleating agent, not the expanded graphite. Just check the result of PM6 and PM14 with degree of crystallinity in the range of 23-23.3%;

> Page 7. Figure 6. Did the authors adopt a single measurement to validate thermal conductivity? Why didn't they do several experiments to get the experimental error?

> Figure 7. What was the total residence time of the material in the internal mixer? Add in manuscript. What type of rotor was used (roller; sigma; cam?)

> Page 7. “Further, they are exfoliated and dispersed under the rotor shear.” There is no way to prove it, considering that the authors did not perform a MET analysis;

> Conclusions. The authors must conclude by showing the importance of the study carried out, as well as the potential application of the reported results;

> In the thermal conductivity results, the authors should explore the obtained values. For example, compare with the literature and present the advantage of the compounds produced.

Round 2

Reviewer 2 Report

In a reanalysis of the manuscript, the authors made significant improvements. In addition, the questions were mostly incorporated in the manuscript. As a consequence, the manuscript has improved quality, and therefore has merit for publication.